# Hyaluronic Acid Conjugated with 17β-Estradiol Effectively Alleviates Estropause-Induced Cognitive Deficits in Rats

**DOI:** 10.3390/ijms242115569

**Published:** 2023-10-25

**Authors:** Mu-Hsuan Chen, Hsiao-Chun Lin, Tzu Chao, Viola Szu-Yuan Lee, Chia-Lung Hou, Tsyr-Jiuan Wang, Jeng-Rung Chen

**Affiliations:** 1Department of Veterinary Medicine, College of Veterinary Medicine, National Chung-Hsing University, No. 145, Xingda Rd., Taichung 402202, Taiwan; sam790418@gmail.com (M.-H.C.); linboni16@gmail.com (H.-C.L.); sue89078@gmail.com (T.C.); 2Basic Research Division, Holy Stone Healthcare Co., Ltd., Taipei 11493, Taiwan; violalee@hshc.com.tw (V.S.-Y.L.); charleshou@hshc.com.tw (C.-L.H.); 3Department of Nursing, National Taichung University of Science and Technology, No. 193, Section 1, Sanmin Rd., Taichung 403027, Taiwan

**Keywords:** cognitive deficit, Alzheimer’s disease, cholinergic septo-hippocampal innervation system, dendritic spine, hippocampus, Morris water maze

## Abstract

Women are at a higher risk of cognitive impairments and Alzheimer’s disease (AD), particularly after the menopause, when the estrous cycle becomes irregular and diminishes. Numerous studies have shown that estrogen deficiency, especially estradiol (E2) deficiency, plays a key role in this phenomenon. Recently, a novel polymeric drug, hyaluronic acid–17β-estradiol conjugate (HA-E2), has been introduced for the delivery of E2 to brain tissues. Studies have indicated that HA-E2 crosses the blood–brain barrier (BBB) and facilitates a prolonged E2 release profile while lowering the risk of estrogen-supplement-related side effects. In this study, we used ovariohysterectomy (OHE) rats, a postmenopausal cognitive deficit model, to explore the effect of a 2-week HA-E2 treatment (210 ng/kg body weight, twice a week) on the cholinergic septo-hippocampal innervation system, synaptic transmission in hippocampal pyramidal neurons and cognitive improvements. Our study revealed an 11% rise in choline acetyltransferase (ChAT) expression in both the medial septal nucleus (MS nucleus) and the hippocampus, along with a 14–18% increase in dendritic spine density in hippocampal pyramidal neurons, following HA-E2 treatment in OHE rats. These enhancements prompted the recovery of cognitive functions such as spatial learning and memory. These findings suggest that HA-E2 may prevent and improve estrogen-deficiency-induced cognitive impairment and AD.

## 1. Introduction

Estrogen is a major component of female gonad hormones, which include estrone, estradiol (E2), estriol and estetrol and are mainly secreted by the ovaries and less so by the liver, adrenal cortex and mammary gland. E2 is not only the most potent endogenous estrogen in regulating physiological female characteristics, but it has also been found to play an extremely important role in neuroprotection in the central nervous system (CNS) and as a neuromodulator that regulates synaptic plasticity and memory [1,2,3,4]. However, when women enter the menopause (between the ages of 45 and 55), estrogen levels in the body drop, and the estrous cycle becomes irregular and begins to taper off. Furthermore, research has also shown that women are twice as likely as men to develop late-onset Alzheimer’s disease (AD), a neurodegenerative disease [5,6]. Postmenopausal estrogen deficiency may contribute to benign cognitive declines and the etiology of AD in women [7]. Younger menopausal women (following surgery, ages lower than 43) tend to experience cognitive decline earlier than women who go through the menopause naturally. Additionally, an earlier age of menopause appears to also be associated with increased neuropathology of AD, especially neuritic plaques [8]. These observations suggest that depletion of E2 in the brain may play an important role in AD development, while retaining normal brain E2 levels can prevent AD progression [9,10,11]. Furthermore, lower estrogen levels in postmenopausal women are associated with the development of dementia [12].

Animal behavior, which encompasses learning, memory formation and cognitive function, often requires the integration of information across different cortical regions, with the hippocampus playing a critical role [13]. Pyramidal neurons in the hippocampus are responsible for the output of integrated information [14], while dendritic spines are widely distributed on the dendritic surface of pyramidal neurons and are believed to be the site of excitatory synaptic connections [15]. Therefore, the number of dendritic spines is closely related to the function of the hippocampus and to adaptive behaviors, learning and memory [2,3,4,16,17,18,19,20]. Our published data indicate that periodic fluctuations in estrogen and estrogen secretion deficiency dynamically alter the spine density in pyramidal neurons of the sensorimotor cortex and the hippocampal CA1 region [2,4]. Such estrogen variations further affect learning and memory formation and the cognitive function of animals [2,4]. In addition, the central cholinergic system plays a crucial role in cognitive functions. Acetylcholine (ACh) is a neurotransmitter synthesized by choline acetyltransferase (ChAT) and degraded by acetylcholinesterase (AChE). ACh’s role in the brain includes regulating excitatory synaptic transmission and participating in cognitive functions such as learning and memory [21]. Studies have indicated that ACh can modulate certain cognitive functions, particularly spatial memory, which relies on the hippocampus [22]. Cholinergic fibers in the hippocampus are mainly projected from the medial septal nucleus (MS nucleus) and regulate the excitatory synaptic transmission of pyramidal neurons [23]. In our previous studies on AD and fetal alcohol spectrum syndrome, we demonstrated that hippocampal function is closely linked to cholinergic septo-hippocampal projection [17,18]. Furthermore, several studies also found that E2 directly affects the performance of the cholinergic system, reducing the high-affinity choline uptake, ChAT activity and ChAT mRNA levels in ovariectomized animals [24,25].

Although exogenous E2 has associated risks of breast cancer and cardiovascular disease [26,27], hormone replacement therapy (HRT) is still widely used to improve the clinical symptoms of menopausal women, such as hot flushes, night sweats, mood swings, vaginal dryness, reduced sexual drive and osteoporosis [28,29]. HRT may also provide patients who have lost estrogen with a strategy to prevent or improve learning, memory loss and cognitive dysfunction. Furthermore, our published studies have shown that morphological and functional defects in the sensorimotor cortex and hippocampal CA1 region caused by estrogen deficiency can be restored by supplementation with E2 or genistein (one of the most abundant isoflavones in soybeans) [3,4]. This indicates that E2 or a similar derivative can modulate learning and memory deficits due to losses in estrogen. Given the risk factors associated with exogenous E2 supplementation, there is a great need for the development of new, safer E2 delivery options for AD.

The delivery of free E2 to the brain could be an important factor in restoring recognition memory. Hyaluronic acid–17β-estradiol conjugate (HA-E2) is a novel hyaluronic acid (HA) drug conjugated with E2. HA-E2 can pass through the blood–brain barrier (BBB) and deliver E2 directly to CNS sites. The novelty of an HA drug conjugate is that the biological polymer HA is the major component of synaptic plasticity and the neuronal extracellular microenvironment. HA biosynthesis is related to the de novo formation of new inhibitory synapses and synaptic remodeling [30]. Recent studies have shown that HA as a drug vehicle can protect E2 and facilitate its entry into the brain; (1) HA as a carrier can help E2 cross the BBB [31], (2) HA-E2 can prolong E2 effects and sustain long-term activity through steric hindrance by the HA polymer to protect E2 from degradation [32], (3) HA conjugated to E2 can protect the 17β-hydroxyl group of E2 from oxidation to estrone, thus reducing the risk of breast cancer and related cardiovascular disease [26,27].

In this study, we examine the effectiveness of HA-E2 treatment in alleviating cognitive deficits in ovariohysterectomy (OHE) rats, as well as its impact on the cholinergic septo-hippocampal innervation pattern and dendritic spine density in hippocampal CA1 pyramidal cells.

## 2. Results

### 2.1. Effect of Exogenous HA-E2 Treatment on Animal Body Weights and Serum E2 Levels in OHE Rats

All rats survived until the end of the study. No treatment-related effect abnormalities, or obvious signs of gross pathological changes, were observed in the study. Body weights and serum E2 levels were determined prior to sacrifice (Figure 1). Body weights in the Cont + Veh, OHE + HA, OHE + E2 and OHE + HA-E2 groups were 283.4 ± 4.7, 313.4 ± 5.1, 329.2 ± 5.0 and 326.8 ± 5.5 g, respectively (Figure 1A, F = 15.29, *p* < 0.001). The body weight of the experimental animals after OHE surgery was significantly higher than the Cont + Veh group regardless of test article treatment (*p* = 0.008, OHE + HA vs. Cont + Veh; *p* < 0.001, OHE + E2 or OHE + HA-E2 vs. Cont + Veh). The serum E2 levels in the Cont + Veh, OHE + HA, OHE + E2 and OHE + HA-E2 groups were 49.2 ± 5.1, 22.3 ± 1.2, 12.9 ± 1.9 and 11.9 ± 2.0 pg/mL, respectively (Figure 1B, F = 35.76, *p* < 0.001). The serum E2 level in all groups was significantly lower than that of the Cont + Veh group after OHE surgery (*p* < 0.001, OHE + HA, OHE + E2 or OHE + HA-E2 vs. Cont + Veh).

### 2.2. Exogenous HA-E2 Treatment for Spatial Learning and Memory in OHE Rats

#### 2.2.1. Escape Latency Test

We assessed spatial learning of the animals treated with HA-E2 via the escape latency test of the Morris water maze (MWM) (Figure 2A and Table 1). Observing the swimming tracks for three consecutive days, all treatment groups showed a reduced escape time and swimming distance with an increasing number of training days, except for the OHE + HA group which was unable to swim directly to the target after entering the pool even on the 3rd day (Figure 2A). There was no significant difference in swimming speed among the groups each day (Table 1). On the 3rd day, there were significant increases in the escape time (*p* = 0.018, OHE + HA vs. Cont + Veh) and swimming distances (*p* = 0.010, OHE + HA vs. Cont + Veh) in the OHE + HA group compared to the Cont + Veh group. On the 3rd day, OHE rats treated with E2 had decreased escape time (*p* = 0.057, OHE + E2 vs. OHE + HA) and swimming distances (*p* = 0.041, OHE + E2 vs. OHE + HA), compared to the OHE + HA group. OHE rats treated with HA-E2 had a significantly decreased escape time (*p* = 0.005, OHE + HA-E2 vs. OHE + HA) and swimming distances (*p* = 0.003, OHE + HA-E2 vs. OHE + HA) compared to OHE + HA rats on the 3rd day.

#### 2.2.2. Spatial Probe Task

The spatial memory of the animals was investigated by the MWM spatial probe test (Figure 2B and Table 2). Tracking of the swimming pattern indicated that OHE + HA rats did not swim a greater distance in the target quadrant (*p* = 0.027 and 0.006 in swimming distance and distance ratio, respectively, OHE + HA vs. Cont + Veh); however, the swimming distance measured in the target quadrant was increased after OHE rats were treated with E2 or HA-E2 compared to the OHE + HA group (*p* = 0.025 and 0.043 in swimming distance and distance ratio, respectively, OHE + E2 vs. OHE + HA; *p* = 0.014 and 0.006 in swimming distance and distance ratio, OHE + HA-E2 vs. OHE + HA). The time that rats spent in a target quadrant was also dependent on the distance traveled, which showed that the swimming time was decreased in OHE + HA rats (*p* = 0.004 and 0.004 in swimming time and time ratio, OHE + HA vs. Cont + Veh), whereas the swimming time in the target quadrant was increased after treatment with E2 or HA-E2 (*p* = 0.009 and 0.008 in swimming time and time ratio, respectively, OHE + E2 vs. OHE + HA; *p* = 0.004 and 0.004 in swimming time and time ratio, respectively, OHE + HA-E2 vs. OHE + HA). Therefore, in OHE rats, the time spent swimming in the target quadrant decreased, but this effect was reversed by treatment with either E2 or HA-E2.

### 2.3. Exogenous HA-E2 Treatment on Cholinergic Septo-Hippocampal Innervation in OHE Rats

Immunohistochemical staining (IHC staining) was used to observe cholinergic septo-hippocampal innervation, containing ChAT+ fiber in the hippocampal CA1 region (Figure 3) and ChAT+ neurons in the MS nucleus (Figure 4). After OHE surgery, the distribution pattern of ChAT+ fiber in the hippocampal CA1 region decreased and could not be restored by HA treatment (Figure 3B). However, in OHE rats receiving E2 or HA-E2 therapy (Figure 3C,D) the region was similar to that of the Cont + Veh group.

Similar results were observed in the cholinergic neuron of the MS nucleus. ChAT expression in the MS nucleus was investigated by the soma area of ChAT+ neurons and their relative integrated optical density (IOD) ratio. The soma area and relative IOD ratio (standardization to the Cont + Veh group) of ChAT+ neurons in the MS nucleus are presented separately in Figure 4E (F = 3.36, *p* = 0.055) and Figure 4F (F = 8.79, *p* = 0.008). The soma area of ChAT+ neurons was significantly decreased in the OHE + HA group (Figure 4E; Cont + Veh: 152.1 ± 4.9 μm^2^/cell, OHE + HA: 140.8 ± 0.6 μm^2^/cell; *p* = 0.043, OHE + HA vs. Cont + Veh), whereas OHE rats treated with either E2 or HA-E2 recovered the area, without significant differences between the E2 and HA-E2 groups (OHE + E2: 148.4 ± 1.2 μm^2^/cell, OHE + HA-E2: 148.9 ± 1.3 μm^2^/cell; *p* = 0.221, OHE + E2 vs. OHE + HA; *p* = 0.184, OHE + HA-E2 vs. OHE + HA). It should be noted that the relative IOD ratio of the ChAT+ neuron of the MS nucleus was decreased in the OHE + HA group. Furthermore, the OHE rats after E2 or HA-E2 treatment could enhance the expression of ChAT (Figure 4F; OHE + HA: 91.0 ± 2.4%, OHE + E2: 99.8 ± 0.5%, OHE + HA-E2: 101.3 ± 2.1%; *p* = 0.022, OHE + E2 vs. OHE + HA; *p* = 0.009, OHE + HA-E2 vs. OHE + HA).

### 2.4. Exogenous HA-E2 Effect on Dendritic Spine of Hippocampal CA1 Pyramidal Neurons

To investigate the relevance of estrogen on synaptic transmission in hippocampal CA1 pyramidal neurons, we used intracellular dye injection to visualize their dendritic structures and analyze changes in spine density (Figure 5). Spine densities on distal apical dendrites and distal basal dendrites of CA1 pyramidal neurons are illustrated in Figure 5B (F = 14.11, *p* < 0.001) and Figure 5D (F = 21.44, *p* < 0.001). Even with HA treatment, E2 deficiency resulted in a 14–17% reduction in the spine density of CA1 pyramidal neurons (Cont + Veh: 18.1 ± 0.3 spines/10 μm in distal apical dendrites and 18.0 ± 0.2 spines/10μm in distal basal dendrites, OHE + HA: 14.9 ± 0.4 spines/10 μm in distal apical dendrites and 15.4 ± 0.3 spines/10 μm in distal basal dendrites; *p* < 0.001 in both dendrites, OHE + HA vs. Cont + Veh). OHE rats after E2 treatment showed a 14–17% increase in the spine density of CA1 pyramidal neurons (OHE + E2: 17.4 ± 0.3 spines/10 μm in distal apical dendrites and 17.5 ± 0.5 spines/10 μm in distal basal dendrites; *p* = 0.002 in distal apical dendrites and *p* < 0.001 in distal basal dendrites, OHE + E2 vs. OHE + HA). Furthermore, HA-E2 treatment also increased the spine density of hippocampal CA1 pyramidal neurons by 14–18% in OHE rats (OHE + HA-E2: 17.7 ± 0.2 spines/10 μm in distal apical dendrites and 17.5 ± 0.2 spines/10 μm in distal basal dendrites; *p* = 0.002 in distal apical dendrites and *p* < 0.001 in distal basal dendrites, OHE + HA-E2 vs. OHE + HA).

## 3. Discussion

This study used OHE surgery to induce estrogen deficiency in a rat model. Following OHE surgery and subsequent survival for 2 weeks, the animals were then treated with exogenous E2 or HA-E2 twice a week for 2 weeks. The main finding of this study was that HA-E2 improves ChAT expression in the MS nucleus, the pattern of ChAT+ fibers in the hippocampal CA1 region and the density of dendritic spines on hippocampal CA1 pyramidal neurons, equivalently to E2. Furthermore, both HA-E2 and E2 treatments enhance the spatial learning and memory deficits of OHE animals in the MWM. Importantly, the equivalent E2 dose of HA-E2 was approximately 60% of that of E2 alone. The HA-E2 conjugate therefore offers the potential to reduce overall E2 dose, thus reducing the risk of other pathologies by two possible mechanisms: (1) HA, as a vehicle, provides a higher BBB penetration rate, which allows HA-E2 to enter the brain more efficiently to exert its effect [31]; (2) HA-E2 can maintain prolonged and sustained activity by protecting E2 from degradation through steric hindrance provided by the HA polymer till it is cleaved slowly from the HA conjugate [32]. Further experimentation is required to deduce the prevalent mechanism.

### 3.1. Exogenous E2 or HA-E2 Treatment Did Not Alter Serum E2 Level or Body Weight

In this study, all rats survived and no abnormalities were observed either after the rats underwent OHE surgery or after they were treated with HA, E2, or HA-E2. After the rats underwent OHE surgery and survived for a further four weeks, it was predictable that endogenous E2 levels in the serum would be depleted (below the measurement limit) or maintained at a very low level. An accelerated increase in body weight was also observed in OHE rats and both of these findings were consistent with results published in the past [3,4,33]. A small amount of E2 could be detected in the serum of some OHE rats, which could be produced by other organs, such as the liver, adrenal cortex and mammary gland. When OHE rats received exogenous E2 or HA-E2 treatment, there were no detectable increases in blood E2 concentration before sacrifice. However, other experimental results in this study showed that E2 or HA-E2 did significantly influence other experimental parameters in OHE rats.

### 3.2. Exogenous E2 or HA-E2 Treatment Modulated Cholinergic Innervation

Many studies have linked E2 to changes in cholinergic neurotransmission in the brain [24,25,34]. A notable point of the study indicated that the mRNA levels of nerve growth factor (NGF) and brain-derived neurotrophic factor (BDNF) appear to be positively correlated with an upregulation or downregulation in the presence or absence of E2 [24,34,35,36]. In this study, we observed that expression of ChAT in cholinergic neurons in the MS nucleus was significantly reduced after estrogen deficiency, decreasing by 9%, whereas these changes could be reversed by treatment with exogenous E2 or HA-E2. Some studies indicate that E2 differentially regulates the expression of neurotrophin receptors (including tropomyosin receptor kinase A (TrkA), TrkB and p75 in basal forebrain cholinergic neurons, mediated via estrogen receptor (ER) α, with an improvement in cholinergic function and survival [24].

ERα is expressed in the cholinergic neurons of the basal forebrain and hippocampus, suggesting that cholinergic transmission might participate in modulating spine plasticity in the hippocampal neuron [3,37,38]. Recent studies also suggested that E2 could increase ACh signaling in the hippocampus by stimulating G protein-coupled estrogen receptor 30 (GPR30) on cholinergic neurons in the basal forebrain [24,39,40]. In our study, IHC staining showed the pattern of ChAT+ fibers in the hippocampal CA1 region was reduced after estrogen deficiency, that these changes could be reversed by treatment with exogenous E2 or HA-E2 and that these changes were consistent with the MS nucleus. Previous research has demonstrated that cholinergic depletion of the forebrain caused a decrease in both the complexity and spine density of CA1 pyramidal neurons and a change in the glutamatergic synaptic transmission of the hippocampus, which in turn leads to impaired hippocampal-dependent learning and memory [17,41,42,43]. Most of the drugs currently approved by the FDA for AD patients, such as donepezil, galantamine, rivastigmine, etc., increase the residence time of ACh in the synaptic cleft, while in this study HA-E2 and E2 treatment (increasing the expression of ChAT) had the same effect. E2 has also been reported to have additional functions that increase the survival of basal forebrain cholinergic neurons [44,45].

### 3.3. Exogenous E2 or HA-E2 Treatment Repopulated the Spine Density of CA1 Pyramidal Neuron

Our previous report demonstrated that estrogen can modulate spine density in layer III and layer V somatosensory cortical pyramidal neurons and in hippocampal CA1 pyramidal neurons. Exogenous E2 restored lost dendritic spines after estrogen deficiency [2,3,4]. Exogenous E2 or HA-E2 significantly increased the spine density of pyramidal neurons in the hippocampal CA1 region of OHE rats, thereby increasing spatial learning ability. Notably, HA-E2 was found to be more effective than the equivalent dose of E2 in maintaining dendritic spines. In addition to the estrogen–cholinergic pathway mentioned above, other possible mechanisms for how estrogen modulates dendritic spines on hippocampal CA1 pyramidal neurons have been proposed. The main ERs on hippocampal pyramidal neurons are ERα and GPR30 [38,46,47]. Expression of ERα in the hippocampus of middle-aged rats appears to decrease while it increases after supplementation with exogenous E2 [48,49,50], although E2 appeared to act directly on pyramidal cells due to the excitatory neurotransmission required for synaptogenesis. Furthermore, research indicated that exogenous E2 also increased the expression of *N*-methyl-D-aspartate receptors (NMDARs) in hippocampal CA1 pyramidal neurons, which can lead the way to the generation of new dendritic spines [46,51]. E2 could also regulate the generation of synaptic structures through processes including activation of NMDARs or metabotropic glutamate receptors (mGluRs), downstream phosphorylation cascades, dynamic changes in postsynaptic membrane expression of a-amino-3-hydroxy-5-methyl-4-isoxazolepropionic acid receptors (AMPARs), cyclic adenosine monophosphate response element binding protein (CREB)-mediated activation of transcription, microtubule-dependent transport of mRNA, local translation of mRNA at the active synapses and dynamic changes in actin cytoskeleton [51,52]. E2 also increased the expression of the CREB/BDNF pathway in the hippocampus and the amygdala particularly [53]. Upregulation of BDNF could activate TrkB receptors and their downstream pathways to increase CREB transcription to modulate synaptic connectivity by increasing the number of synaptic structures [54,55]. These factors are beneficial in increasing the spine density of pyramidal neurons. Some studies also suggest that the biological polymer HA may play a role in the formation and stability of inhibitory synapses and synapse remodeling [30,56]. HA receptor CD44 plays a crucial role in promoting HA retention and inhibiting excitatory synapse formation by regulating Rho GTPase signaling. As a result, an increase in the level of HA between neurons results in a decrease in excitatory synapse formation, an increase in inhibitory synapse formation and inhibition of action potential formation. Recently, HA has been extensively studied as a drug delivery vehicle for neurotrophins, including BDNF and NGF, in the treatment of CNS injuries and diseases, and has demonstrated remarkable efficacy [57,58,59]. However, studies that have investigated the use of HA treatment alone have not yielded sufficient benefits to animal models of neurological diseases. In our OHE rat model, HA treatment alone did not improve cognitive deficits and failed to increase the dendritic spines (excitatory synapses) on hippocampal pyramidal neurons.

Based on our results, we have shown that estrogen deficiency leads to a significant decrease in cholinergic neuron function in the basal forebrain nucleus, which can be reversed by exogenous E2 or HA-E2 treatment. Restoration of cholinergic neuron function (similar to the mechanism of most FDA-approved Alzheimer’s drugs) increases ACh levels between neurons in the brain, promotes excitatory synaptic transmission between pyramidal neurons and improves animal cognitive deficits [60]. Assuredly, we cannot rule out the possibility that E2 or HA-E2 directly acts on the ER in hippocampal pyramidal neurons to regulate dendritic spine formation. Furthermore, Picciotto’s study has shown that the function of cholinergic neurons is also closely related to hippocampal synaptic plasticity. ACh regulates hippocampal synaptic plasticity through the activation of downstream signaling pathways via muscarinic acetylcholine receptors (mAChRs) and nicotinic acetylcholine receptors (nAChRs) [61].

### 3.4. Limitations of the Present Study

Previous research has confirmed that hormonal fluctuations, including gonadal hormones, have an impact on cognitive function [62]. For example, hormonal changes in pregnant women can affect tasks related to frontal lobe and hippocampal function, including attention and memory [62]. Our current research has primarily employed behavioral and morphological approaches to investigate estrogen hormone imbalance and its relation to cognitive function. However, our current evidence cannot confirm the following: (1) the direct impact of the disappearance or supplementation of female hormones on the cellular functions in the hippocampus and MS nucleus; (2) whether estrogen supplementation-mediated changes in ACh affect synaptic plasticity in the hippocampus. Therefore, more research is still needed to explore the current hypotheses further by the following research methods and strategies: (1) examine the distribution pattern and quantitative changes in ERs in the hippocampus and MS nucleus following OHE surgery or exogenous E2 supplementation; (2) detect changes in receptors (nAChR and mAChR) or downstream signaling pathways associated with ACh-mediated synaptic plasticity by employing relevant assays. The implementation of these research strategies and methods can contribute to strengthening the current hypothesis and advancing our understanding of the relationship between estrogen imbalance and cognitive function. The development of HA-E2 primarily aims at treating cognitive impairments. However, it was evident that utilizing only the OHE-induced cognitive deficit animal model for testing is insufficient. Therefore, the utilization and testing of various dementia models with cognitive deficits will be required in the future. Additionally, several questions that remain unanswered regarding the use of HA-E2 for the treatment of cognitive impairments: (1) whether HA-E2 administration results in a higher BBB penetration rate compared to E2 and the amount of HA-E2 that acts on the hippocampus and MS nucleus; (2) whether HA-E2 can reduce common pathological features of dementia, such as tau protein or β-amyloid; (3) whether long-term use of HA-E2 increases the risk of cardiovascular disease and breast cancer. To address the above questions, we considered the following potential methods and strategies for future investigation: (1) utilize positron emission tomography (PET) to monitor real-time changes in E2 or HA-E2 levels within the hippocampus and MS nucleus; (2) measure tau protein and β-amyloid levels after HA-E2 treatment; (3) conduct long-term observation of factors related to cardiovascular disease and breast cancer after HA-E2 treatment. Finally, our dosing frequency has been based on the estrous cycle of rats to confirm the therapeutic effects of HA-E2 in cognitive deficit rats. However, to maximize the efficiency of HA-E2 treatment in the future, considerations can include changes to the HA-E2 formulation, reducing the dose, extending the duration of drug impact or adapting it to less invasive administration routes. These adjustments aim to make future use of HA-E2 more patient-friendly, improve quality of life and reduce potential risks associated with E2 treatment.

## 4. Materials and Methods

### 4.1. Animals

Seventy-five female Sprague Dawley rats, about 200 g in body weight, were obtained from BioLASCO Taiwan Co., Ltd. All procedures for the care and feeding of experimental animals were in accordance with the Animal Care and Use Committee of the National Chung-Hsing University. All animals were individually caged (43 × 23 × 20 cm^3^) at constant temperature (24 ± 1 °C) and humidity (60 ± 5%) in a light-controlled room (12 h light–dark cycle) and were provided with food and water ad libitum. Sprague Dawley rats without surgery and treated with vehicle were the control group (Cont + Veh, n = 17). The surgical protocol used for OHE rats was described previously [3,4]. The OHE surgical rats were divided into 3 groups: OHE rats treated with HA (OHE + HA, n = 11), OHE rats treated with E2 (OHE + E2, n = 23), and OHE rats treated with HA-E2 (OHE + HA-E2, n = 24), respectively. In each experimental batch, a random selection of one normal animal belonging to the Cont + Veh group was made and used as a reference baseline for the other animals in different groups within that batch. The presence of this reference animal allows for comparison and contrast with the experimental results of other groups, ensuring the reliability and consistency of the experiments. The administration of each treatment is described below. In this study, all animals, regardless of whether they underwent OHE surgery or received article treatments, survived until the end of the experiment. Before sacrifice, vaginal smears were performed on the experimental animals to determine their estrous cycles and confirm that the control animals were in the proestrus stage.

### 4.2. Experiment Schedule

Figure 6 shows the experimental schedule. The selected rats underwent the OHE surgery on day 0 (P0). Rats were assigned to groups and treated with either vehicle control, HA, E2 or HA-E2 on days P17, P20, P24 and P27 according to their group. The latency training trial of the MWM was started on P24 and continued for three days. The probe trial of the MWM was carried out on day P28. At the end of the experiment, all of the animals were sacrificed for histological and morphological analysis.

### 4.3. Drug Administration

HA (molecular weight 360 kDa, Bloomage Biotechnology, Jinan, China) and HA-E2 were provided by Holy Stone Healthcare (Taipei, Taiwan). The structure of HA-E2 is illustrated in Figure 7 as confirmed by ^1^H NMR spectrum. Preliminary proof-of-concept studies were conducted based on previously published references [63]. To determine the effective E2 dosage and the effectiveness of E2 treatment regimens, doses of 280 and 350 ng/kg were evaluated and the latter showed a significant therapeutic effect. Similarly, we evaluated HA-E2 treatment regimens at doses of 140, 210, 280, and 350 ng/kg and found that the dose of 210 ng/kg began to have a significant therapeutic effect. Test articles were prepared as follows. HA, 2.1 mg was dissolved in 20 mL 1× phosphate-buffered saline (PBS) and stirred for 3 h at room temperature as the stock and then diluted with 1× PBS to a concentration of 52.5 µg/mL. 17β-estradiol, 2 mg (E2, Sigma-Aldrich, Inc., Louis, MO, USA) was dissolved in 1 mL absolute ethanol (Sigma-Aldrich, Inc., Louis, MO, USA) and then diluted to 100 times in PBS as stock. Furthermore, E2 stock was diluted with 1× PBS (Sigma-Aldrich, Inc., Louis, MO, USA) to a concentration of 350 ng/mL. HA-E2, 2 mg was dissolved in 2 mL 1× PBS and stirred for 3 h at room temperature as stock and then diluted with 1× PBS to a concentration of 52.5 µg/mL. Animals were treated with the same volume (1 mL/kg) of vehicle (1× PBS), HA, E2 (350 ng/kg body weight) or HA-E2 (210 ng/kg E2) according to the schedule defined in Figure 6 by intravenous injection via the tail vein at 9 a.m. on the day of treatment. Administration of the test articles to OHE animals was performed in a single-blind manner without the operator’s knowledge of the drug type.

### 4.4. MWM Task

Behavioral tests were conducted at 10 a.m. on test days. A modified MWM task was used to assess the spatial learning and memory of the animals [3,17,18]. The maze consisted of a black circular pool 145 cm in diameter and 23 cm deep. A round transparent platform 20 cm in diameter was placed 3 cm under the water. One visual cue (cardboard star) was located at the edge of the pool, which was on the opposite side of the platform. Throughout the entire MWM task process, the operators were required to wear white laboratory attire. Furthermore, after placing the animals into the water maze, operators promptly returned to standardized positions to ensure the consistency of the testing environment. The MWM task was divided into an escape latency test and the spatial probe test. Animal behavioral performances during the escape latency test and spatial probe test were recorded with a video camera. The recorded videos were then analyzed with the SMART video tracking system (SMART 3.0V, Panlab, Harvard Apparatus, Cambridge, UK).

#### 4.4.1. Escape Latency Test

To assess escape latency, rats were tested with two trials per day for 3 consecutive days. To ensure fairness in each of the two daily trials, animals were placed in two different quadrants of the pool. Each animal was oriented towards the walls of the pool upon entry and the entry positions were consistent for each individual. This was carried out to ensure that the test results were not influenced by the initial positions and to allow the animals to learn and adapt to different environmental conditions. Additionally, during the 3 consecutive days of testing, the entry positions of the animals were not the same as the previous day, eliminating the potential effects of fixed strategies. This approach aimed to more accurately assess their learning and spatial navigation abilities. Rats were allowed to remain on the platform for 60 s if they located it (escaped) within 180 s or were placed on the platform for 60 s if they failed to locate the underwater platform within 180 s. A recovery period of 10 min was allowed between the two trials conducted each day. Data from the two daily trials of the escape latency test, including escape latency (swimming time), swimming distance and swimming speed, were averaged for subsequent analysis.

#### 4.4.2. Spatial Probe Test

To conduct the spatial probe test, we utilized the same pool and environment as the escape latency test except for the platform being removed from the pool. A fan-shaped virtual target quadrant was defined according to the previous position of the platform. After the final escape latency task and one day of rest, all rats were placed in the opposite quadrant of the target quadrant facing the wall of the pool. The rats were allowed to swim once for 30 s, and the swimming path, including swimming distance and time spent in the target quadrant, was analyzed.

### 4.5. Sacrifice and Preparation of Serum and Brain Tissues

Rats were weighed and deeply anesthetized with 7% chloral hydrate and 0.02% xylazine (5 mL/kg body weight). For the preparation of serum, blood samples were collected following cardiac puncture in heparinized microtube solution to prevent blood clotting (lithium heparin, BD Vacutainer). Samples were immediately centrifuged at 2880× *g* for 20 min and serum was separated. Serum samples were sent to Union Reference Laboratory (Taichung, Taiwan) to measure E2 levels (the detected limit of estrogen was 11.8 pg/mL. When the detected value was less than the limit, it was recorded as 0 pg/mL).

Tissue preparation for intracellular dye injection and IHC staining was described previously [2,3,4,17,18]. Briefly, rats were transcardially perfused with 2% paraformaldehyde (PFA) in 0.1 M phosphate buffer (PB) for 30 min. Brains were carefully removed and sectioned with a vibratome (Technical Products International, St. Louis, MO, USA) into three parts: (1) 2000 µm thick coronal slices containing MS nucleus for IHC staining; (2) two pieces of 350 μm thick coronal slices containing hippocampus for intracellular dye injection and (3) 2000 μm thick coronal slices containing hippocampus for IHC staining.

### 4.6. IHC Staining

The 2000 µm thick coronal slices for IHC staining were postfixed in 4% PFA (in 0.1 M PB) for 1 day. Postfixed thick slices were cryoprotected by 30% sucrose (in 0.1 M PB) and divided into 30 µm thick serial sections with a cryostat (Leica Biosystems Division of Leica Microsystems, Inc., Deer Park, IL, USA) for subsequent IHC staining. The cryosections were processed for IHC staining for ChAT antibody to reveal the distribution of ChAT+ neurons in the MS nucleus and ChAT+ fiber in the hippocampus. During the entire process of IHC, all the animal slices from the same batch, after completing the group marking, were pooled. These slices were collectively transferred to the next staining container at each subsequent staining step to ensure consistent staining conditions within each batch. The sections were first incubated with 1% H_2_O_2_ (Kento Chemical, Co., Inc., Tokyo, Japan) and 0.2% Triton X-100 (Fisher Scientific, Inc., Hampton, NH, USA) in 0.1 M PB for 1 h to remove endogenous peroxidase activity. Then, the sections were reacted with goat anti-ChAT (1:500, Merck Millipore, Burlington, MA, USA) for 18 h at 4 °C. Biotinylated rabbit anti-goat (1:200, Vector Laboratories, Inc., Newark, CA, USA) immunoglobulin was used as the secondary antibody for 1 h at room temperature. They were then incubated with standard avidin–biotin HRP reagent (Vector Laboratories, Inc., Newark, CA, USA) for 1 h at room temperature. They were reacted with 0.05% 3,3′-diaminobenzidine (DAB, Sigma-Aldrich, Inc., St. Louis, MO, USA) and 0.01% H_2_O_2_ in 0.05 M Tris buffer. Reacted sections were mounted on slides, air-dried and coverslipped with Permount (Fisher Scientific, Inc., USA). We observed and captured images of four slices containing the hippocampus in each rat using a 40× objective lens. Six serial sections containing MS nucleus per rat were observed and images were captured using a 4× objective lens, and the ChAT+ soma area and ChAT+ IOD value (OD ×area) were measured using Image-Pro Plus 6.0 (Media Cybernetics, Inc., Rockville, MD, USA). We configured the filter range in Image-Pro Plus 6.0 (area range of 50–300 μm^2^) to select all ChAT+ neurons within the MS nucleus of the aforementioned slices. This allowed us to obtain the average soma area and IOD value for each ChAT+ neuron in the MS nucleus. Due to the presence of one Cont + Veh animal in each experimental batch as a reference baseline, we standardized the ChAT+ IOD values of the other groups within the same batch to ensure the comparability of results for each experimental batch.

### 4.7. Intracellular Dye Injection and Immunoconversion

Methods of intracellular dye injection and immunoconversion followed previously published reports [2,3,4,17,18]. The 350 μm thick coronal slices for intracellular dye injection were treated with 10^−7^ M 4′,6-diamidino-2-phenyl-indole (DAPI, Sigma-Aldrich, Inc., St. Louis, MO, USA) in 0.1 M PB to label nuclei in the hippocampal CA1 region and were then placed on the stage of a fixed-stage fluorescence microscope (Olympus, Tokyo, Japan). A 20× long-working distance objective lens under filter set (390–420 nm, FT 425, LP 450) was used to monitor the hippocampal CA1 pyramidal layer. A 4% Lucifer yellow (LY) solution (diluted with water) was added to a glass micropipette. A three-axial hydraulic micromanipulator (Narishige, Tokyo, Japan) was used to move the glass micropipette and an intracellular amplifier (Axoclamp–IIB, Axon, Foster city, CA, USA) was used to release LY into the selected neuron via generating 3 min of negative current. Subsequently, the slices were postfixed in 4% PFA (in 0.1 M PB) for 1 day. The slices were then cryoprotected by 30% sucrose (in 0.1 M PB) and sectioned into 60 µm thick serial sections with a cryostat.

The previous serial sections were first incubated with 1% H_2_O_2_ and 1% Triton X-100 in 0.1 M PB for 30 min to remove endogenous peroxidase activity and then incubated with biotinylated rabbit anti-LY (1:200, Thermo Fisher Scientific, Inc., Waltham, MA, USA) in PBS containing 2% bovine serum albumin (Sigma-Aldrich, Inc., St. Louis, MO, USA) for 18 h at 4 °C and then incubated with standard avidin–biotin HRP reagent for 1 h at room temperature. Subsequently, they were reacted with 0.05% DAB and 0.01% H_2_O_2_ in 0.05 M Tris buffer. Finally, reacted sections were mounted on slides, air-dried and coverslipped with Permount. To determine the density of dendritic spines in the hippocampal CA1 region, distal apical and distal basal dendrites of pyramidal neurons were analyzed. Five independent cells of the hippocampal CA1 region and three segments from each dendrite were observed by a 100× objective lens and the spine density per 10 μm was randomly counted.

### 4.8. Statistical Analysis

IBM SPSS Statistics 20 (International Business Machines, Co., Armonk, NY, USA) was used for statistical analysis. In order to confirm whether the data conform to normality, all data were observed for their distribution and subjected to the Shapiro–Wilk normality test. The statistical significance of the escape latency test and spatial probe test of the MWM was determined using Kruskal–Wallis one-way analysis of variance (ANOVA) with Dunn’s post hoc multiple comparisons to find the differences between the groups. The results were expressed as the median (interquartile range, IQR = Q1–Q3). The statistical significance of the body weights, E2 levels in serum, swimming distance in the spatial probe test of the MWM, soma area and relative IOD ratio of ChAT+ neurons in the MS nucleus and spine density of hippocampal CA1 pyramidal neurons were determined using one-way ANOVA with Tukey’s post hoc multiple comparisons to determine the differences between the groups and the data were expressed as mean ± SEM. *p* < 0.05 was considered significantly different.

## 5. Conclusions

E2 therapy has been repeatedly shown to alleviate many symptoms caused by hypogonadism in recent years and there is no doubt about its effectiveness. This study confirmed that a lower equivalent dose of HA-E2 had the same level of effectiveness as a higher equivalent E2 treatment in improving estrogen-deficiency-induced decreased cholinergic septo-hippocampal innervation, spine loss in CA1 pyramidal neurons and spatial learning and memory deficit. Our novel compound design, using hyaluronic acid as a carrier, not only allows E2 to be released more precisely and lastingly in the brain, but also reduces the total amount of E2 that is required to be administered, thereby reducing the risk of other E2-related diseases. Therefore, HA-E2 has great potential as a new drug for estrogen-deficiency-induced cognitive impairment and Alzheimer’s disease.

## Figures and Tables

**Figure 1 ijms-24-15569-f001:**
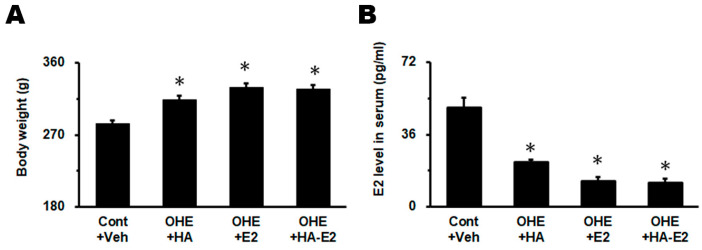
**Change in body weights and estradiol (E2) levels in serum post treatments.** Body weights and E2 levels in serum are analyzed in (**A**) and (**B**), respectively. The animal numbers were as follows: 17 in Cont + Veh, 11 in OHE + HA, 23 in OHE + E2 and 24 in OHE + HA-E2. *, *p* < 0.05 between the marked groups and Cont + Veh.

**Figure 2 ijms-24-15569-f002:**
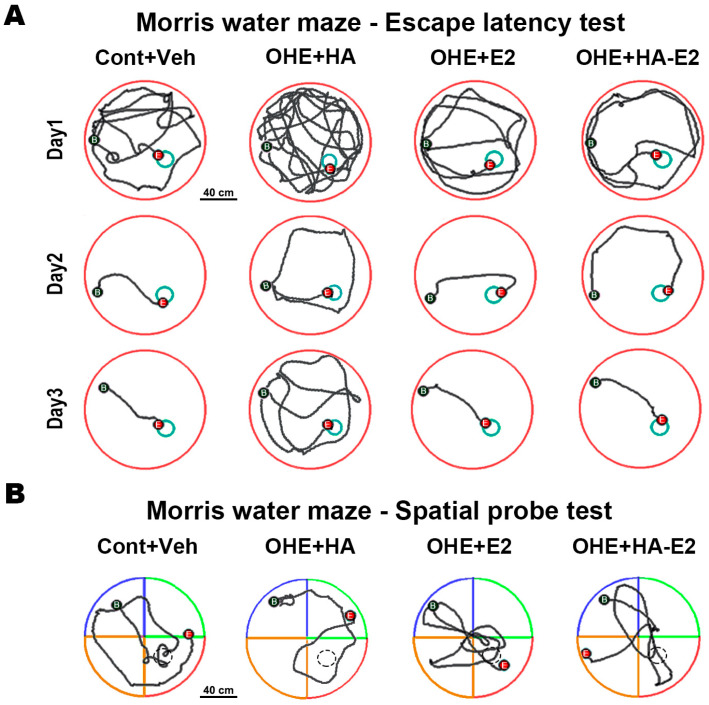
**The swimming tracks of Morris water maze (MWM) tasks.** The swimming tracks of the MWM escape latency test for three consecutive days and spatial probe test are shown in (**A**,**B**). The red area represents the target quadrant, while the green, orange and blue areas represent non-target quadrants in (**B**). Bar = 40 cm in (**A**,**B**).

**Figure 3 ijms-24-15569-f003:**
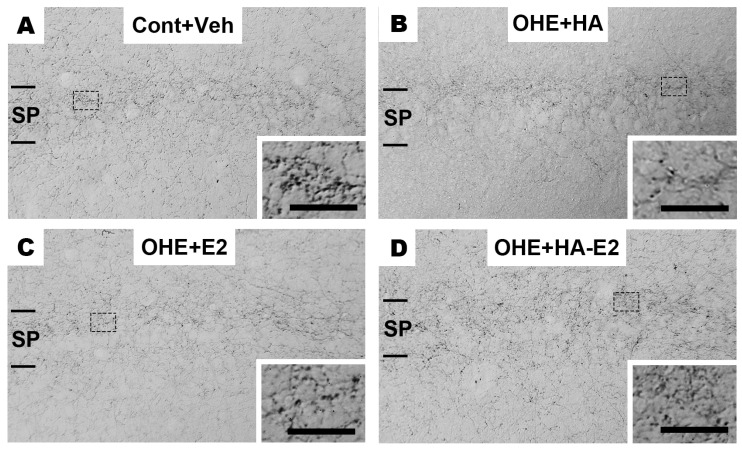
**The distribution pattern of choline acetyltransferase (ChAT)+ fibers in the hippocampal CA1 region.** Micrographs of the ChAT+ fibers in hippocampal CA1 region of the four groups are illustrated in (**A**–**D**). The inset shows a higher magnification view of each micrograph. SP, stratum pyramidale. Bar = 100 µm for all micrographs; 25 µm for insets.

**Figure 4 ijms-24-15569-f004:**
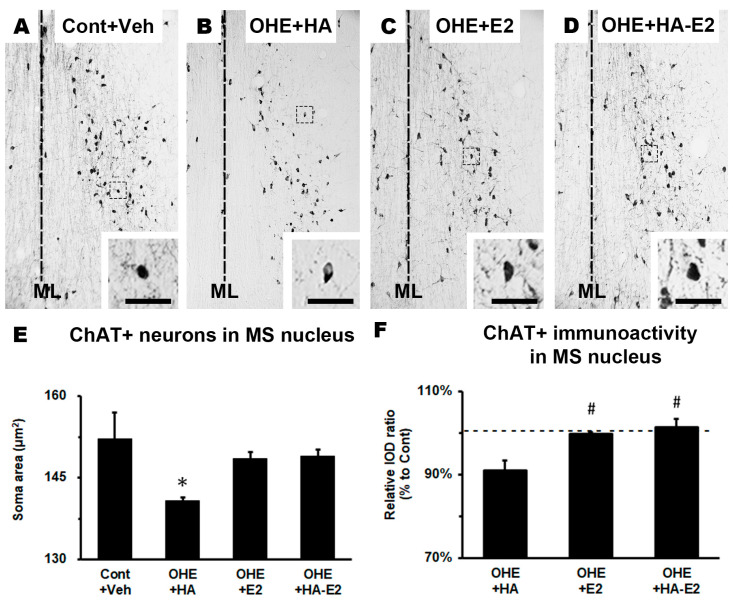
**The expression of ChAT in the medial septal nucleus** (**MS nucleus).** Micrographs of the ChAT+ neurons in MS nucleus of the four groups are illustrated in (**A**–**D**). The soma area and relative integrated optical density (IOD) ratio of ChAT+ neurons in the MS nucleus are analyzed in (**E**,**F**). The number of animals in each group was 4. *, *p* < 0.05 between the marked groups and Cont + Veh; #, *p* < 0.05 between the marked groups and OHE + HA. ML, midline. Bar = 200 µm for all micrographs; 50 µm for insets. The dotted lines in F indicate 100%.

**Figure 5 ijms-24-15569-f005:**
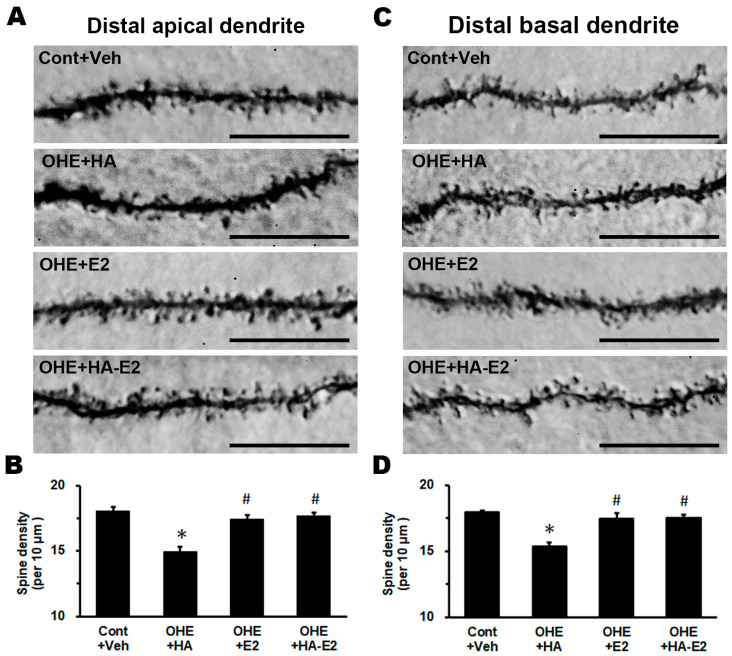
**The spine density of hippocampal CA1 pyramidal neurons.** Representative micrographs of the distal apical dendrites (**A**) and distal basal dendrites (**C**) of the hippocampal CA1 pyramidal neurons from each group are illustrated. The number of animals in each group was 4. Spine density per 10 μm of distal apical and distal basal dendrites is analyzed and plotted individually in (**B**,**D**). *, *p* < 0.05 between the marked groups and Cont + Veh; #, *p* < 0.05 between the marked groups and OHE + HA. Bar = 10 µm for all micrographs.

**Figure 6 ijms-24-15569-f006:**
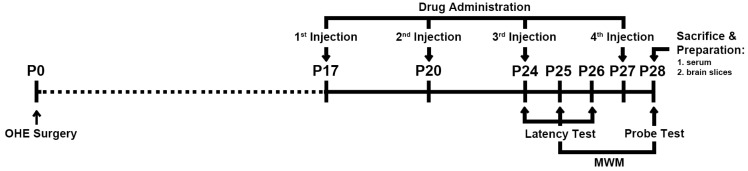
**Experiment design.** Experimental schedule for ovariohysterectomy (OHE) surgery, test article administration and behavioral assessment. The OHE surgery was performed on day 0 (P0). MWM, Morris water maze.

**Figure 7 ijms-24-15569-f007:**
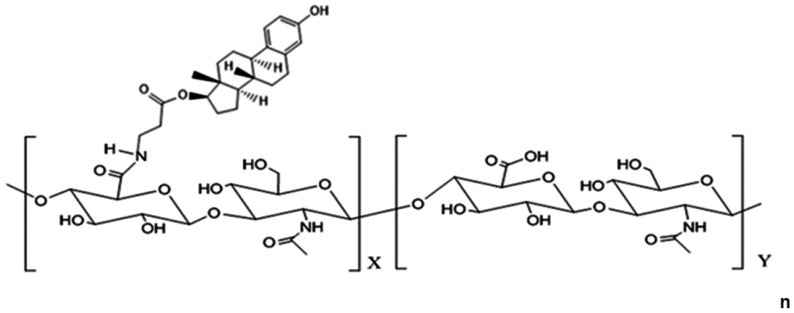
**Structure of hyaluronic acid–17β-estradiol conjugate (HA-E2).** HA-E2 is a conjugate of the biological polymer hyaluronic acid (HA) in its sodium salt form, sodium hyaluronate and E2. X = number of hyaluronate carboxyl groups substituted with estradiol; Y = number of unsubstituted hyaluronate disaccharides.

**Table 1 ijms-24-15569-t001:** Behavioral performance of the Morris water maze (MWM) escape latency test.

	Groups			
	Con + Veh	OHE + HA	OHE + E2	OHE + HA-E2
Animal numbers	17	11	23	24
Escape latency (seconds)				
Day1 (*p* = 0.172)	62.9 (IQR = 41.4–77.5)	60.1 (IQR = 25.3–78.6)	40.5 (IQR = 22.3–73.9)	34.0 (IQR = 20.2–58.6)
Day2 (*p* = 0.210)	8.7 (IQR = 5.2–15.3)	9.4 (IQR = 7.2–22.9)	16.5 (IQR = 6.0–22.2)	9.6 (IQR = 6.2–27.1)
Day3 (*p* = 0.007)	6.6 (IQR = 5.0–10.1)	11.6 (IQR = 10.9–21.9) *	7.7 (IQR = 5.1–10.8)	6.0 (IQR = 4.4–9.4) ^#^
Swimming distance (meters)				
Day1 (*p* = 0.170)	11.6 (IQR = 8.5–14.0)	11.5 (IQR = 4.9–16.2)	8.2 (IQR = 4.4–13.4)	6.8 (IQR = 4.4–11.7)
Day2 (*p* = 0.257)	1.6 (IQR = 1.0–2.5)	2.4 (IQR = 1.4–5.5)	3.7 (IQR = 1.2–4.9)	2.0 (IQR = 1.3–5.4)
Day3 (*p* = 0.004)	1.2 (IQR = 0.9–1.7)	2.9 (IQR = 2.0–4.1) *	1.5 (IQR = 1.0–2.4) ^#^	1.2 (IQR = 0.9–1.7) ^#^
Swimming speed (cm/s)				
Day1 (*p* = 0.811)	19.2 (IQR = 16.8–21.2)	19.6 (IQR = 18.9–21.7)	20.0 (IQR = 17.9–21.1)	19.4 (IQR = 17.7–21.9)
Day2 (*p* = 0.711)	21.9 (IQR = 18.5–23.9)	22.2 (IQR = 19.3–25.4)	20.0 (IQR = 18.2–24.3)	21.0 (IQR = 21.0–22.9)
Day3 (*p* = 0.728)	19.2 (IQR = 16.8–23.2)	20.4 (IQR = 18.5–23.0)	20.9 (IQR = 18.3–24.5)	19.3 (IQR = 17.4–23.6)

Kruskal–Wallis test followed by Dunn test: *, *p* < 0.05 between the marked groups and Cont + Veh. ^#^, *p* < 0.05 between the marked groups and OHE + HA.

**Table 2 ijms-24-15569-t002:** Behavioral performance of the MWM spatial probe test.

	Groups			
	Con + Veh	OHE + HA	OHE + E2	OHE + HA-E2
Animal numbers	17	11	23	24
Swimming distance in the target quadrant				
Value (meter) (*p* = 0.011)	1.8 (IQR = 1.6–2.3)	1.3 (IQR = 0.8–1.4) *	1.9 (IQR = 1.3–2.4) ^#^	1.9 (IQR = 1.4–2.3) ^#^
Ratio (%) (*p* = 0.006)	31.1 (IQR = 28.3–41.4)	23.8 (IQR = 17.0–25.9) *	31.3 (IQR = 21.1–39.8) ^#^	32.3 (IQR = 24.0–39.4) ^#^
Swimming time in the target quadrant				
Value (second) (*p* = 0.002)	9.4 (IQR = 8.0–13.1)	5.6 (IQR = 3.9–8.0) *	9.6 (IQR = 6.4–13.5) ^#^	10.1 (IQR = 7.2–12.9) ^#^
Ratio (%) (*p* = 0.002)	31.2 (IQR = 26.7–43.6)	18.7 (IQR = 13.1–26.8) *	31.9 (IQR = 21.3–45.0) ^#^	33.7 (IQR = 24.1–43.1) ^#^

Kruskal–Wallis test followed by Dunn test: *, *p* < 0.05 between the marked groups and Cont + Veh. ^#^, *p* < 0.05 between the marked groups and OHE + HA.

## Data Availability

The data presented in this study are available on request from the corresponding author. The data are not publicly available due to our ongoing research, which is still in progress and must be completed before it can be made public.

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
