# Peer review of "Hyaluronic Acid Conjugated with 17β-Estradiol Effectively Alleviates Estropause-Induced Cognitive Deficits in Rats"

_ijms, 2023, doi:10.3390/ijms242115569_

Round 1

Reviewer 1 Report

This is an interesting study with a well designed methodology.

I only have one suggestion/comment:

In figure 5 and 6, the images are very difficult to discern. If possible, please provide high quality immunofluorescence images instead.

Additional comments:

- Please provide further details in the methods section:  How was the scoring performed? Were the raters blinded? Expand on the staining strategies and the counting/quantification.

- A native english speaker should revise this paper to make it more readable. 

- Expand on the limitations. What further experiments are required and what are the limitations of this study?

Minor improvements are needed.

Reviewer 2 Report

The Paper entitled “Hyaluronic acid-conjugated 17β-estradiol provide more effectively to alleviate estropause-induced cognitive deficits in rats by Mu-Hsuan Chen and co-authors highlighted the neuroprotective effects of Hyaluronic acid-conjugated 17β-estradiol in estropause-induced cognitive dysfunction and other pathological manifestations. The overall findings are based on behavioral and biochemical alterations.

My humble comments and suggestions may be found in the given lines.

1.       The English style is poor, as may be seen in the title and the whole abstract. The abstract has been weakly designed, without proper format.

The end of the introduction says, “In this study, we investigate the efficacy of HA-E2 treatment for cognitive deficit in 98 OHE rats, the changes in the pattern of cholinergic septo-hippocampal innervation and spine 99 density in hippocampal CA1 pyramidal cells”, which needs careful consideration.

In the introduction, the authors have given that “Animal behavioral performance related to learning and memory formation…. Which may be changed accordingly.

In the discussion, the authors have shown that “Based on our results, we aim to show that…. Need correction.

2.       In the abstract, the authors haven’t added any information about the dosing and schedule of treatment.

3.       No information on the methods used for the behavioral analyses. How was it conducted? Were the experimenters blind to the treatment groups? If not, how the business was reduced?

4.       In the material and methods sections, the authors have given details of the animal used. What was the basis for this selection, and grouping? Any scientific reason? How about the death of the rats, if some rats have died with the surgery?

5.       Was the escape latency a separate test, or was just conducted in the same MWM apparatus?

6.       The same is true with the Spatial working memory test. The author may read more about the MWM test and its subcategories.

7.       In Figure 1, the authors have mentioned the first drug, second drug, and so on… How many drugs you have given? Or you have just given different doses.

8.       I am just wondering why the administration time is quite short, in such a short time it prevented the AD-like phenotype.

9.       No information about the experimenters has been added, were they blind to the assigned groups during the performing of the behavioral studies?

10.   In the introduction the information related to the role of acetylcholine in cognitive functions hasn’t been presented.

11.   The provided images are of Golgi staining so far, the methods used to conduct such staining haven’t been presented here. I am more interested in the measurements of this staining.

12.   There are several research gaps and shortcomings in the given paper, but the authors haven’t discussed what is really missing and what may be performed by the other scientists to strengthen the current hypothesis related to Hormonal imbalance and cognitive functions.

13.   Much effort is required to make it publishable.

14.   Good luck

Weak
